# Breakdown of self-incompatibility due to genetic interaction between a specific *S*-allele and an unlinked modifier

Yan Li [1,2] ✉, Ekaterina Mamonova[2], Nadja Köhler[2], Mark van Kleunen [2,3] & Marc Stift [2] ✉

Breakdown of self-incompatibility has frequently been attributed to loss-of-function mutations of alleles at the locus responsible for recognition of self-pollen (i.e. the *S*-locus). However, other potential causes have rarely been tested. Here, we show that self-compatibility of $S_1S_1$-homozygotes in selfing populations of the otherwise self-incompatible *Arabidopsis lyrata* is not due to *S*-locus mutation. Between-breeding-system cross-progeny are self-compatible if they combine $S_1$ from the self-compatible cross-partner with recessive $S_1$ from the self-incompatible cross-partner, but self-incompatible with dominant *S*-alleles. Because $S_1S_1$ homozygotes in outcrossing populations are self-incompatible, mutation of $S_1$ cannot explain self-compatibility in $S_1S_1$ cross-progeny. This supports the hypothesis that an $S_1$-specific modifier unlinked to the *S*-locus causes self-compatibility by functionally disrupting $S_1$. Self-compatibility in $S_{19}S_{19}$ homozygotes may also be caused by an $S_{19}$-specific modifier, but we cannot rule out a loss-of-function mutation of $S_{19}$. Taken together, our findings indicate that breakdown of self-incompatibility is possible without disruptive mutations at the *S*-locus.

To avoid the negative consequences of self-fertilization, about half of the angiosperms have some form of self-incompatibility[1,2]. For example, Brassicaceae have a sporophytic self-incompatibility system, which renders plants self-incompatible through recognition and rejection of self-pollen. Two tightly linked recognition genes (the female gene *S*-locus Receptor Kinase – *SRK*, and the male gene *S*-locus Cystein Rich – *SCR*) encode stigma- and pollen proteins, respectively, together forming what is commonly referred to as the *S*-locus[3]. If stigma- and pollen proteins have matching specificities (as would be the case with self-pollination), pollen tubes cannot grow, thus preventing fertilization[4–6]. Self-incompatibility frequently breaks down[7,8], and many extant self-compatible and selfing species have evolved from self-incompatible ancestors[9–14]. Since the loss of self-

incompatibility is the first step towards the evolution of selfing, it is of particular interest to understand its underlying genetic basis.

Disruptive mutations at the self-recognition locus (*S*-locus) have likely caused the breakdown of self-incompatibility that ultimately gave rise to the selfing species *Arabidopsis thaliana*[15], *Capsella rubella*[16] and *C. orientalis*[17], and selfing lineages of Siberian *A. lyrata*[18]. The most convincing evidence for a primary role of the *S*-locus came from a gain-of-function study that transferred a functional *S*-locus from self-incompatible *A. lyrata* into self-compatible *A. thaliana*, which led to the complete restoration of self-incompatibility in a few accessions[15,19,20]. In other accessions, the transformation had no effect or only led to transient self-incompatibility, indicating that these accessions must have carried loss-of-function mutations both at the

[1]Key Laboratory of Ecosystem Network Observation and Modeling, Institute of Geographic Sciences and Natural Resources Research, Chinese Academy of Sciences, Beijing 100101, China. [2]Ecology, Department of Biology, University of Konstanz, Universitätsstraße 10, D-78457 Konstanz, Germany. [3]Zhejiang Provincial Key Laboratory of Plant Evolutionary Ecology and Conservation, Taizhou University, Taizhou 318000, China. ✉e-mail: yan.li@uni-konstanz.de; marc.stift@uni-konstanz.de

*S*-locus and for additional genetic elements required for functional self-incompatibility[19]. The documentation of a non-functional *S*-locus in several extant selfing species (*C. rubella*[16], *C. orientalis*[17], *A. thaliana*[15,20], *A. kamchatica*[21], *Leavenworthia alabamica*[22]) provided support for a primary role of the *S*-locus in the breakdown of self-incompatibility. In *C. orientalis*, for example, a disruptive mutation at the *SCR* gene was inferred to underlie the breakdown of self-incompatibility[17]. Several other genetic mechanisms could theoretically explain the loss of self-incompatibility, including disruptive mutations at any of the genes encoding downstream components of self-incompatibility[13], or secondary loci encoding factors that interact with the function of self-recognition genes or their downstream components[23–27]. Indeed, modifiers have been invoked to explain the variability in the successful restoration of self-incompatibility by the transformation of *A. thaliana*[26]. However, to the best of our knowledge, there are no empirical examples showing conclusive evidence for mechanisms other than loss-of-function mutations at the *S*-locus to be responsible for the breakdown of self-incompatibility.

North American *A. lyrata* (subspecies *lyrata*) has recently emerged as a potential case where an alternative mechanism has led to a breakdown of self-incompatibility. The species is mainly self-incompatible, but several populations have become self-compatible and highly inbreeding[28]. Based on associations with specific *S*-alleles, there appear to be multiple origins of selfing. Two selfing populations consist of homozygotes for *S*-allele 1[29] ($S_1$, recessive to all other *S*-alleles[30]), while three populations consist of homozygotes for *S*-allele 19 ($S_{19}$, which belong to a higher dominance class[30,31]). One selfing population is mixed, with homozygotes for $S_1$, $S_{19}$, and $S_{27}$[29]. The related self-compatible species *A. arenicola* may have been derived from one of these[14,32]. The presence of self-incompatible $S_1$ homozygotes in outcrossing populations suggested that loss-of-function mutations in self-recognition genes of the $S_1$ haplotype could not explain the loss of self-incompatibility for this specificity. Instead, it was hypothesized that a modifier-locus unlinked to the *S*-locus causes self-compatibility in $S_1$-homozygotes of *A. lyrata*[29]. However, the evidence for this modifier hypothesis is limited, and there is still no direct evidence for a functional link between self-compatibility and the $S_1$ and $S_{19}$ haplotypes.

Here, we show that the breakdown of self-incompatibility in selfing populations of *Arabidopsis lyrata* is functionally linked to $S_1$ and $S_{19}$. We do this based on intra- and inter-population crosses between self-compatible and self-incompatible plants from six outcrossing (self-incompatible) and six selfing (self-compatible) populations. By *S*-locus genotyping a subset of the cross-progeny, we further show that the functional link between $S_1$ and self-compatibility is not due to a loss-of-function mutation of $S_1$. Instead, we infer that a modifier unlinked to the *S*-locus disrupts the function of $S_1$ and confers self-compatibility in cross-progeny homozygous for $S_1$. Whether a similar mechanism functionally disrupts $S_{19}$ remains to be tested.

## Results

### Self-compatible and self-incompatible progeny emerge from between-breeding-system crosses

We tested whether progeny from crosses between between-breeding system (BBS), i.e. crosses between self-compatible (SC) and self-incompatible (SI) plants, would show variation in the breeding system. To quantify the breeding system, we performed manual self-pollinations on progeny, measured the resulting fruits and calculated a fruit-length-based index (the SC-index). The range of SC-index values of progeny from such crosses (BBS$_{♀SI × ♂SC}$ and BBS$_{♀SC × ♂SI}$) included the complete spectrum from completely SI (SC-index < 0.25) to completely SC (SC-index > 0.75). Although on average intermediate to the SC-index of progeny from crosses within breeding systems (BP$_{♀SC × ♂SC}$ and BP$_{♀SI × ♂SI}$; $z = −1.10$, $p = 0.87$; no significant effect of C5 in Supplementary Table 1), most between-breeding-system progeny could be

phenotyped as either SC (SC-index > 0.75; 450 out of 904) or SI (SC-index < 0.25; 363 out of 904) (Supplementary Table 2). Consequently, the SC-index showed a bimodal distribution with a median of 0.74 and peaks at c. 0 and c. 1.2 (Fig. 1). The remaining 91 progeny had intermediate SC-index values and could thus not be phenotyped unambiguously according to our (conservative) thresholds (Supplementary Table 2). The cross direction (i.e. whether the maternal or paternal

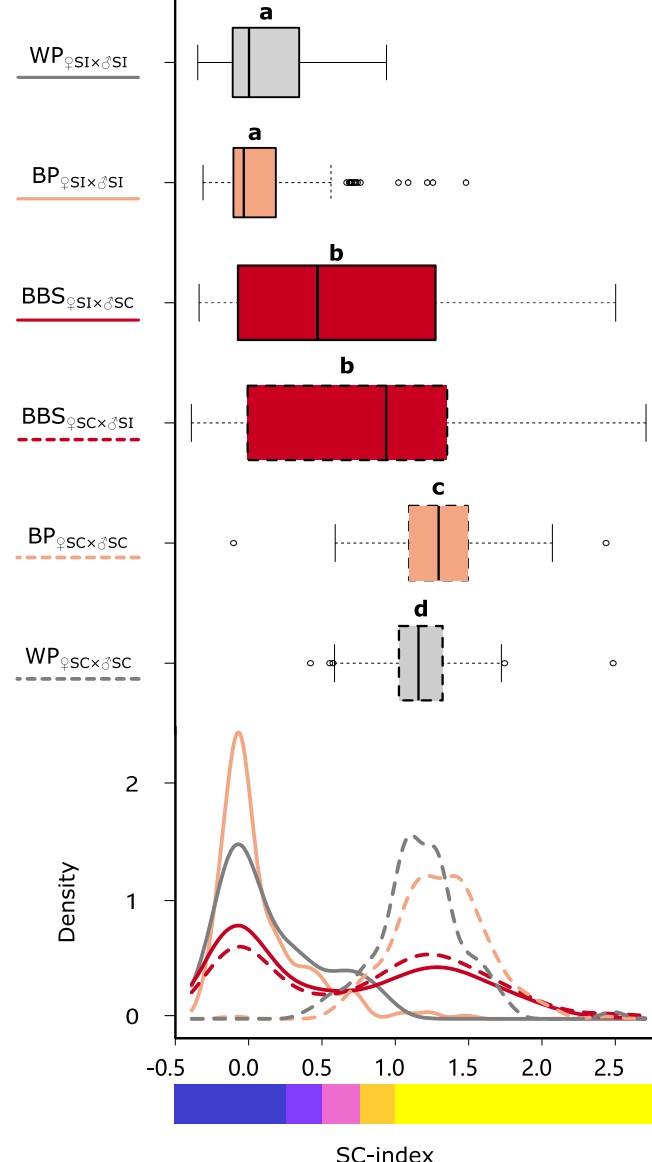

**Fig. 1 | SC-index values for progeny from crosses within and between breeding systems.** Boxes delimit the interquartile range (IQR) with the median indicated as a solid line. Whiskers extend to 1.5×IQR or to the lowest/highest data point within 1.5×IQR. Points beyond the whiskers' limits represent outliers. Means of boxes marked with different letters are significantly different based on post-hoc two-sided z-tests (cf. Supplementary Table 1; WP$_{♀SI × ♂SI}$: $N = 75$; BP$_{♀SI × ♂SI}$: $N = 210$; BBS$_{♀SI × ♂SC}$: $N = 455$; BBS$_{♀SC × ♂SI}$: $N = 449$; BP$_{♀SC × ♂SC}$: $N = 230$; WP$_{♀SC × ♂SC}$: $N = 84$ biologically independent plants). Abbreviations: WP (grey): within-population; BP (orange): between-population; BBS (red): between-breeding-system (by design also between populations). Solid borders and lines are used for cross-types with self-incompatible maternal parents (♀SI) and dashed borders and lines for cross-types with self-compatible maternal parents (♀SC). Note the bimodal shape for the BBS cross-type with peaks matching those observed for the cross-types from between-SI-populations crosses and between-SC-populations crosses. Source data are provided as a Source Data file.

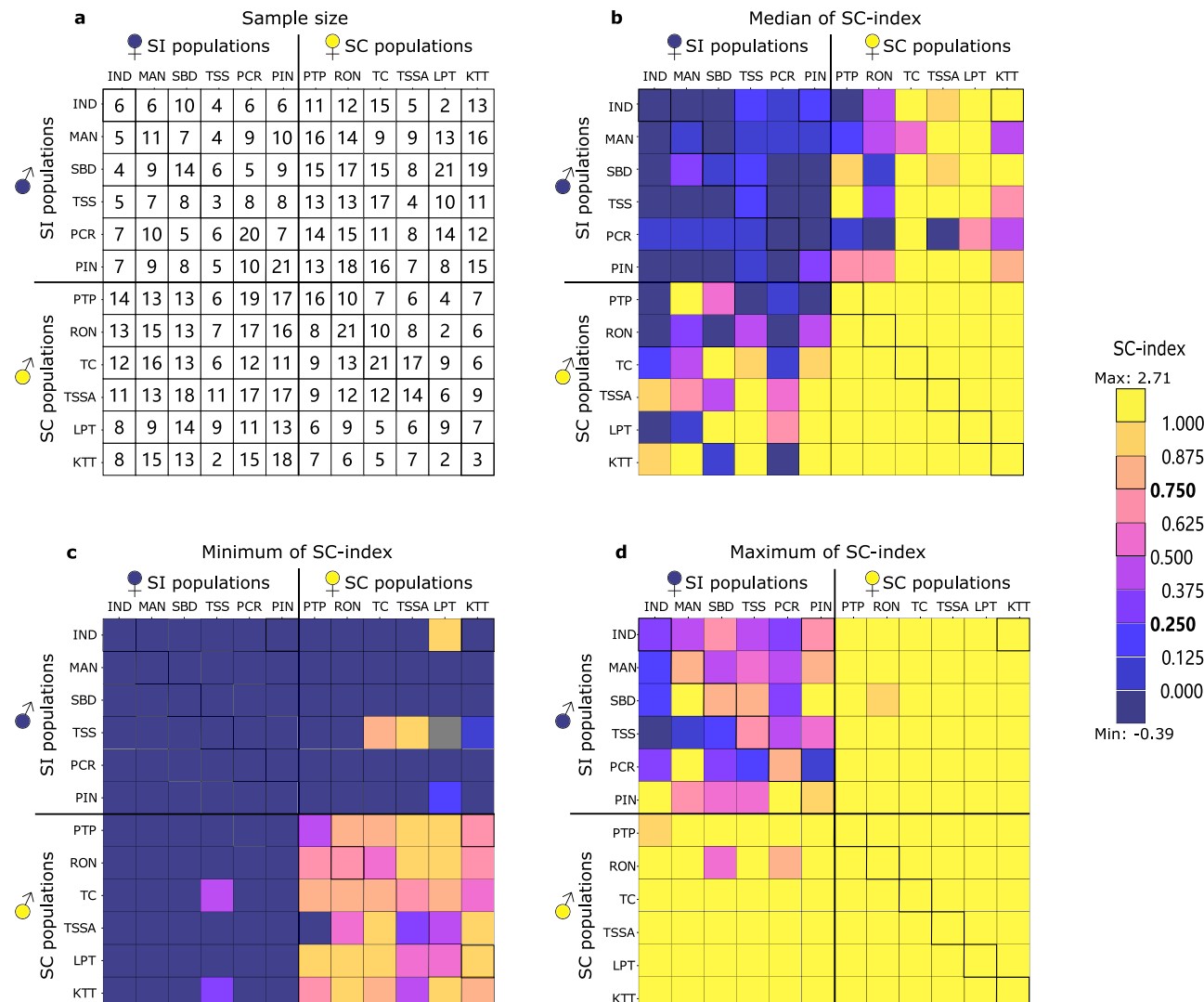

**Fig. 2 | Summary statistics for the SC-index for within- and between-population crosses. a** Cell-specific number of progeny for which the breeding system phenotype was determined; **b** Heat-map of median SC-index; **c** Heat-map of minimum SC-index; **d** Heat-map of maximum SC-index. Population codes correspond to the ones in Supplementary Table 3. Diagonal cells marked with an extra outline represent progeny from within-population crosses. Other squares represent each possible combination of populations in two directions (with the maternal population shown at the top). The 36-cell top-left quadrat and 36-cell bottom-right quadrat represent crosses between plants with the same breeding system (♀SI × ♂SI and ♀SC × ♂SC, respectively). The 36-cell top-right quadrat and 36-cell bottom-left quadrat represent crosses between plants with different breeding systems (♀SC × ♂SI and ♀SI × ♂SC, respectively). Source data are provided as a Source Data file.

cross-partner was SC) did not have a significant effect on the average SC-index value of progeny from between-breeding-system crosses ($z = -2.08$, $p = 0.26$; no significant effect of C4 in Supplementary Table 1; Fig. 1), although the proportion of SC plants was higher when the mother was SC (Supplementary Table 2). The patterns were not specific to any cross-combination, as the vast majority of population combinations (30 out of 36) resulted in both SI and SC progeny (Fig. 2; compare panels c and d). At the seed-family level, 93 out of 256 families segregated for breeding system, i.e., they contained both SI progeny and SC progeny (Supplementary Fig. 1). These breeding-system phenotypes depended on the S-haplotypes inherited from the SC and SI cross partners.

### Cross-progeny mostly SC if inheriting $S_1$ from SC and SI cross-partner

A total of 12 SC cross partners were homozygous $S_1S_1$ (B80$_{H50}$B80$_{H50}$), and originated from the selfing populations RON ($N = 6$) and PTP

($N = 6$) (Supplementary Data 1). ♀SC × ♂SI and ♀SI × ♂SC crosses with these plants yielded 333 progeny that could be phenotyped for the breeding system: 106 were SC, 182 were SI and 45 intermediate (22 SC-index value between 0.25 and 0.5; 23 SC-index value between 0.5 and 0.75) (Supplementary Table 2). Of the 106 SC progeny, we genotyped the S-locus of 48, which had all inherited B80 haplotypes associated with $S_1$ (H43, H62, H115, H130) from the SI cross-partner (Fig. 3a, b). Of the 182 SI progeny, we genotyped the S-locus of 62, which had inherited nine different haplotypes. Five SI progeny had inherited a haplotype associated with most recessive allele $S_1$ (H43 or H62) from the SI cross-partner, but the remaining 57 had inherited haplotypes known to be associated with S-alleles putatively dominant to $S_1$ (H48 with $S_3$; H75 with $S_{19}$; H128 with $S_{39}$; H55, H82, H92, and H129 with unknown S-alleles) (Fig. 3a, b). Of the 45 progeny with intermediate SC-index values, we genotyped the S-locus of 22, which had inherited seven haplotypes (H43 and H62 with $S_1$; H48 with $S_3$; H75 with $S_{19}$; H128 with $S_{39}$; H55 and H82 with unknown S-alleles) (Fig. 3a, b).

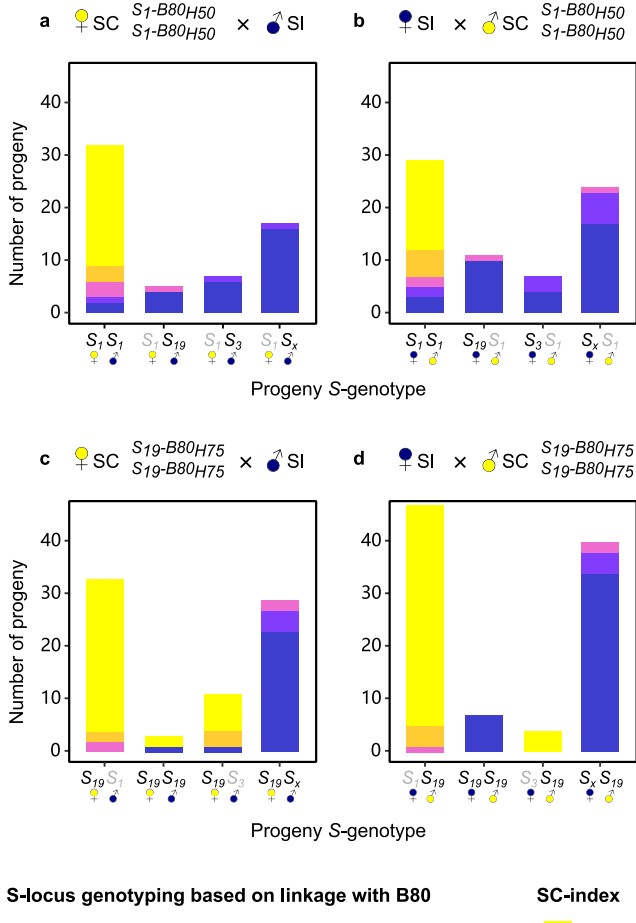

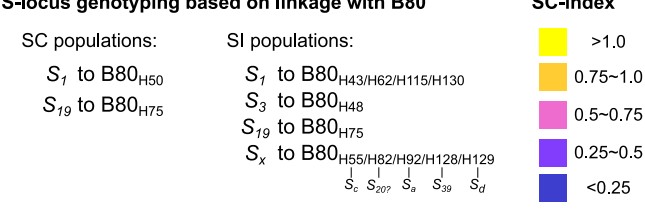

**Fig. 3 | Progeny SC-index by *S*-genotype for between breeding system (BBS) crosses (♀SI × ♂SC and ♀SC × ♂SI).** Panel **a** and **b** show progeny from crosses with maternal (**a**) or paternal (**b**) SC partners homozygous for $S_1$ associated with B80$_{H50}$. Panel **c** and **d** show progeny from crosses with maternal (**c**) or paternal (**d**) SC partners homozygous for $S_{19}$ associated with B80$_{H75}$. *S*-genotypes were inferred by known linkage to B80 sequences as indicated in the figure (also see Supplementary Data 1). $S_1$ and $S_3$ are recessive to all other known *S*-alleles, and $S_3$ is dominant over $S_1$. Putatively expressed *S*-alleles are given in black, nonexpressed (recessive) *S*-alleles in grey. $S_x$ represents several *S*-alleles different from $S_1$, $S_3$, or $S_{19}$ (see legend for details). Source data are provided as a Source Data file.

## Cross-progeny mostly SC if inheriting dominant $S_{19}$ from SC cross partner and recessive $S_1$ or $S_3$ from SI cross partner

A total of 20 SC cross partners were homozygous $S_{19}S_{19}$ (B80$_{H75}$B80$_{H75}$), and originated from the selfing populations KTT ($N = 6$), LPT ($N = 6$), TC ($N = 6$) and TSSA ($N = 2$) (Supplementary Data 1). ♀SC × ♂SI and ♀SI × ♂SC crosses with these plants yielded 498 progeny that could be phenotyped for the breeding system: 320 were SC, 141 were SI and 37 intermediate (23 SC-index value between 0.5 and 0.75; 14 SC-index value between 0.25 and 0.5) (Supplementary Table 2). Of the 320 SC progeny, we genotyped the *S*-locus of 93, which had inherited seven different haplotypes. Of those, 76 had inherited B80 haplotypes associated with the most recessive allele $S_1$ (H43, H62, H115, H130) and 14 the B80 haplotype associated with the second-most recessive allele $S_3$ (H48) from the SI cross-partner (Fig. 3c, d). Of the remaining three, two had inherited H75, likely the result of inadvertent

self-pollination of the (maternal) SC cross-partner. One had inherited a B80 haplotype from SI parent PCR17. Of the 141 SI progeny, we genotyped the *S*-locus of 67, which had inherited eight different haplotypes. One of the 67 SI progeny had inherited the haplotype associated with the second-most recessive allele $S_3$ (H48), but the remaining 66 had inherited haplotypes associated with *S*-alleles putatively dominant to $S_1$ and $S_3$ (H75 with $S_{19}$; H128 with $S_{39}$; H55, H82, H92, H127 and H129 with unknown *S*-alleles) (Fig. 3c, d). Of the 37 progeny with intermediate SC-index values, we genotyped the *S*-locus of 15, which had inherited four haplotypes (H43 with $S_1$; H128 with $S_{39}$; H55 and H82 with unknown *S*-alleles) (Fig. 3c, d).

## ♀SI × ♂SI crosses give a low frequency of SC progeny

For crosses within SI populations (WP$_{♀SI × ♂SI}$), the SC-index of progeny had a unimodal distribution with a median of 0.003. The majority of progeny (51 out of 75) had an SC-index below the threshold of 0.25, and was thus phenotyped as SI (Fig. 1, Supplementary Table 2). However, a considerable number of progeny (5 out of 75) had an SC-index above 0.75 and was thus phenotyped as SC. The remaining 19 progeny had intermediate SC-index values (Supplementary Table 2). Similar patterns emerged for crosses between SI populations (BP$_{♀SI × ♂SI}$, $z = −1.63$, $p = 0.53$; no significant effect of C1 in Supplementary Table 1; Fig. 1). With a median SC-index of −0.03 (Fig. 1), 168 out of 210 progeny were phenotyped as SI, 7 as SC and the remaining 35 as intermediate (Supplementary Table 2). Thus, overall, most progeny from crosses between SI parents were SI (Fig. 2), but 5 SC progeny emerged after crosses within four different SI populations (MAN, PCR, PIN, SBD), and a further 7 after crosses between SI populations (seven different population-combinations; Fig. 2d). All 12 SC progeny from crosses between SI parents involved parents with at least on copy of $S_1$ (Supplementary Fig. 2).

## ♀SC × ♂SC crosses give mostly SC progeny

For crosses within SC populations (WP$_{♀SC × ♂SC}$), the SC-index of progeny had a unimodal distribution with a median of 1.15. The vast majority of progeny (78 out of 84) had an SC-index above the conservative threshold of 0.75, and were thus phenotyped as SC, the remaining 6 were intermediate (Fig. 1, Supplementary Table 2). A similar pattern emerged for crosses between SC populations (BP$_{♀SC × ♂SC}$), with an even higher SC-index median of 1.30 ($z = 4.05$, $p < 0.001$; significant effect of C2 in Supplementary Table 1, Fig. 1), and with 222 out of 230 progeny unambiguously phenotyped as SC, one as SI and 7 intermediate (Supplementary Table 2). Non-SC progeny did not emerge more frequently if crosses involved SC partners from different *S*-locus backgrounds (Supplementary Fig. 3). Thus, taken together, there was no evidence for restoration of self-incompatibility.

## Discussion

North American *A. lyrata* is usually self-incompatible, but at least two independent transitions to self-compatibility and selfing have occurred[28], putatively functionally linked to two specific *S*-alleles ($S_1$ and $S_{19}$)[29]. We determined the breeding system of more than 1,503 progeny from crosses within and between selfing and outcrossing populations by calculating an SC-index, which quantifies fruit-length after self-pollination. This provided unique insights in the genetic basis of the loss of self-incompatibility in North American *A. lyrata*. Our key finding was that crosses between self-compatible (SC) and self-incompatible (SI) plants (between-breeding-system crosses: BBS$_{♀SI × ♂SC}$ and BBS$_{♀SC × ♂SI}$) yielded both SI and SC progeny. Focusing on BBS crosses involving SC plants from the two most common *S*-locus backgrounds associated with selfing, $S_1$ (linked to B80$_{H50}$) and $S_{19}$ (linked to B80$_{H75}$), we could further show that the progeny's breeding system depended on the *S*-alleles inherited from the SI partner, confirming the functional link between self-compatibility and both the $S_1$ and $S_{19}$ haplotypes. Such a link could

arise if loss-of-function mutations of self-recognition genes at the *S*-locus caused the breakdown of self-incompatibility, as was shown for the ancestors of the extant selfing species *A. thaliana*[15], *C. rubella*[16] and *C. orientalis*[17]. Below, we will discuss that our findings are compatible with a similar scenario in the $S_{19}$-B80$_{H75}$ background, but cannot explain the breakdown of self-incompatibility in the $S_1$-B80$_{H50}$ background of North American *A. lyrata*. Instead, our findings provide strong support for the hypothesis[29] that self-compatibility in the $S_1$ background is mediated by a modifier unlinked to the *S*-locus.

In crosses involving SC cross partners homozygous for $S_1$ (linked to B80$_{H50}$), and SI partners (with different *S*-alleles), SC only emerged in progeny homozygous for $S_1$. In other words, progeny could be SC only if they inherited $S_1$ from both the SC partner ($S_1$ linked to B80$_{H50}$) and the SI cross-partner ($S_1$ linked to other B80 variants, e.g., B80$_{H43}$). Owing to its recessivity to all other *S*-alleles[30], $S_1$ is the most frequent *S*-allele in natural outcrossing populations and $S_1S_1$ homozygotes are common (e.g. Ref. 29,31) and normally SI, with a very low frequency of SC individuals[28]. Reflecting this, in our sample of 36 SI cross-parents, 26 parents had at least one copy of $S_1$ (Supplementary Data 1). Natural $S_1S_1$ homozygotes are only consistently SC in selfing populations fixed for $S_1$-B80$_{H50}$, which does not occur in outcrossing populations[29]. This provides a strong indication that $S_1$ is a functional *S*-allele, and that the observed association of self-compatibility with $S_1S_1$ homozygotes is not due to a loss-of-function mutation shared by all $S_1$-variants. A loss-of-function-mutation specific to $S_1$-B80$_{H50}$ could still explain the association of self-compatibility with $S_1$-B80$_{H50}$ homozygotes in natural populations. However, this could only explain our finding of SC in $S_1S_1$ progeny (which only have one $S_1$-B80$_{H50}$ copy), if $S_1$-B80$_{H50}$ acts dominantly over other intact $S_1$ counterparts such as $S_1$-B80$_{H43}$. Since loss-of-function mutations are usually recessive[33], this scenario is highly unlikely.

A dominantly acting modifier suppressing the function of $S_1$ would provide a more plausible explanation that our crosses between homozygous $S_1$-B80$_{H50}$ SC plants and SI plants only yielded SC plants if the SI partner contributed $S_1$. Moreover, modifier-action could explain the rare emergence of SC $S_1S_1$ individuals in natural outcrossing populations[28] and in our SI × SI crosses (Supplementary Fig. 2), which cannot be explained by a loss-of-function mutation of $S_1$-B80$_{H50}$, since $S_1$-B80$_{H50}$ is absent from outcrossing populations[29]. Modifiers interacting with the *S*-locus are common features of homomorphic self-incompatibility systems in general[23–25], and specifically in the Brassicaceae[13,26,27]. In *A. halleri*, *S*-locus-linked modifiers mediate the dominance hierarchy between *S*-alleles[34] (reviewed in ref. 35). For example, $S_1$ (AhS1 in *A. halleri*; AlS1 in *A. lyrata*) is recessive to all other *S*-alleles. This is mediated by an *S*-linked precursor gene (found in all *S*-alleles, except $S_1$), which encodes an sRNA that inactivates $S_1$[34]. Like *S*-locus dominance modifiers, the proposed $S_1$-specific modifier that confers self-compatibility in $S_1$ homozygotes also inactivates $S_1$, but is not physically linked to the *S*-locus. Maintenance of modifier-alleles conferring partial or complete self-compatibility is expected as a means to provide reproductive assurance under conditions where costs of complete self-incompatibility are high[36–38]. The proposed $S_1$-specific modifier is thus likely maintained at low frequency in outcrossing populations, but has been driven to fixation (along with $S_1$-B80$_{H50}$) during the transition to a selfing mating system in the RON and PTP populations. Taken together, our findings provide strong support for the modifier hypothesis[29] that self-compatibility in $S_1S_1$ homozygotes is conferred by an $S_1$-specific modifier.

For haplotype $S_{19}$, our data is less conclusive, but confirms the functional link between $S_{19}$-B80$_{H75}$ from selfing populations and the breakdown of self-incompatibility and provides several useful insights. First, the function of $S_{19}$ appears to be disrupted without affecting its dominance over the two most recessive *S*-alleles ($S_1$ and $S_3$). Maintenance of *S*-allele dominance after loss-of-function mutation in *SCR* was also found in *C. orientalis*[17] and Siberian *A. lyrata*[18]. Second, if an

*S*-locus mutation underlies the functional link between $S_{19}$ and self-compatibility, this mutation must only be present in the $S_{19}$-B80$_{H75}$ haplotype found in SC populations, but not the $S_{19}$-B80$_{H75}$ haplotype found in SI populations. Owing to its dominance level, the $S_{19}$-haplotype is relatively rare (10%) in SI populations[29]. Accordingly, only four (out of 36) SI parents had a single copy of $S_{19}$-B80$_{H75}$ (Supplementary Data 1). If this copy had a loss-of-function mutation, crosses with other SI plants homozygous for the recessive $S_1$ should have produced equal frequencies of SC progeny (progeny that inherited $S_{19}$-B80$_{H75}$ and thus became $S_1S_{19}$) and SI progeny (progeny that did not inherit $S_{19}$-B80$_{H75}$ and thus became $S_1S_x$). However, such crosses yielded exclusively non-SC progeny (in total 12 progeny). This speaks against a general loss-of-function mutation of the $S_{19}$-B80$_{H75}$ haplotype. This would resemble the scenario proposed for the Siberian selfing lineages of *A. lyrata*, where the dominant *S*-allele AhS12 (named $S_{42}$ or AlS42 in *A. lyrata*) carried loss-of-function mutations in selfing, but not in outcrossing lineages[18]. We conclude that our findings are compatible with a loss-of-function mutation specific to the $S_{19}$-B80$_{H75}$ variant from SC populations, but cannot rule out that functional suppression of this variant is caused by a dominant modifier allele (inherited from the SC cross-partner).

Although most progeny could be phenotyped unambiguously as self-incompatible or self-compatible, substantial numbers of progeny had intermediate phenotypes. Theoretically, our SC-index could be affected by inbreeding depression leading to reduced seed set in individuals that are in principle self-compatible. However, the overwhelming bimodality in the SC-index values in the F$_1$ progeny, both overall (Fig. 1) and within families (Supplementary Fig. 1) does not match the unimodal distributions expected if inbreeding depression was the main driver of reduced seed set after selfing. This suggests that progeny with intermediate SC-index values represent cases of leaky self-incompatibility or so-called pseudo-self-fertility, which has been described for several other species with gametophytic and sporophytic self-incompatibility[24,39–42]. It may be maintained in self-incompatible populations as a means of reproductive assurance[38], but the mechanistic and genetic basis of leaky self-incompatibility is still poorly understood.

In conclusion, our findings confirm that the association between $S_1$ and $S_{19}$ and self-compatibility has a functional basis. Furthermore, for $S_1$, the functional association is due to a genetic interaction with a modifier unlinked to the *S*-locus. The latter documents a breakdown of self-incompatibility that cannot be explained by loss-of-function mutations at the *S*-locus. Future work should consider whether such a scenario is unique to North American *A. lyrata*, or may also explain the breakdown of self-incompatibility in other systems.

## Methods

### Source plant material and crossing design

Loss of self-incompatibility and transition to high selfing rates had been documented in six *A. lyrata* populations[28,43]. The timing of this transition is unknown, but the lack of a selfing syndrome[44] and limited purging of genetic load[45,46] suggest a relatively recent origin. To study the inheritance of self-compatibility, we used seed material from 12 North American *A. lyrata* populations (kindly provided by Barbara Mable, University of Glasgow) with contrasting breeding and mating systems. According to genotyping of progeny arrays and self-pollinations for these 12 populations (Supplementary Table 3), six populations (hereafter SI populations) mainly consist of self-incompatible (SI) plants, and have multi-locus outcrossing rates over 80%. SI populations all display *S*-allele diversity[29]. The remaining six populations (hereafter SC populations) are considered to be selfing, with high frequencies of self-compatible (SC) plants (four populations 100%, one 88% and one 50%), and much lower outcrossing rates (five of them no more than 31%, one population [TSSA] 41%; see Supplementary Table 3)[28,29]. One of these (TSSA) clusters with the nearby SI

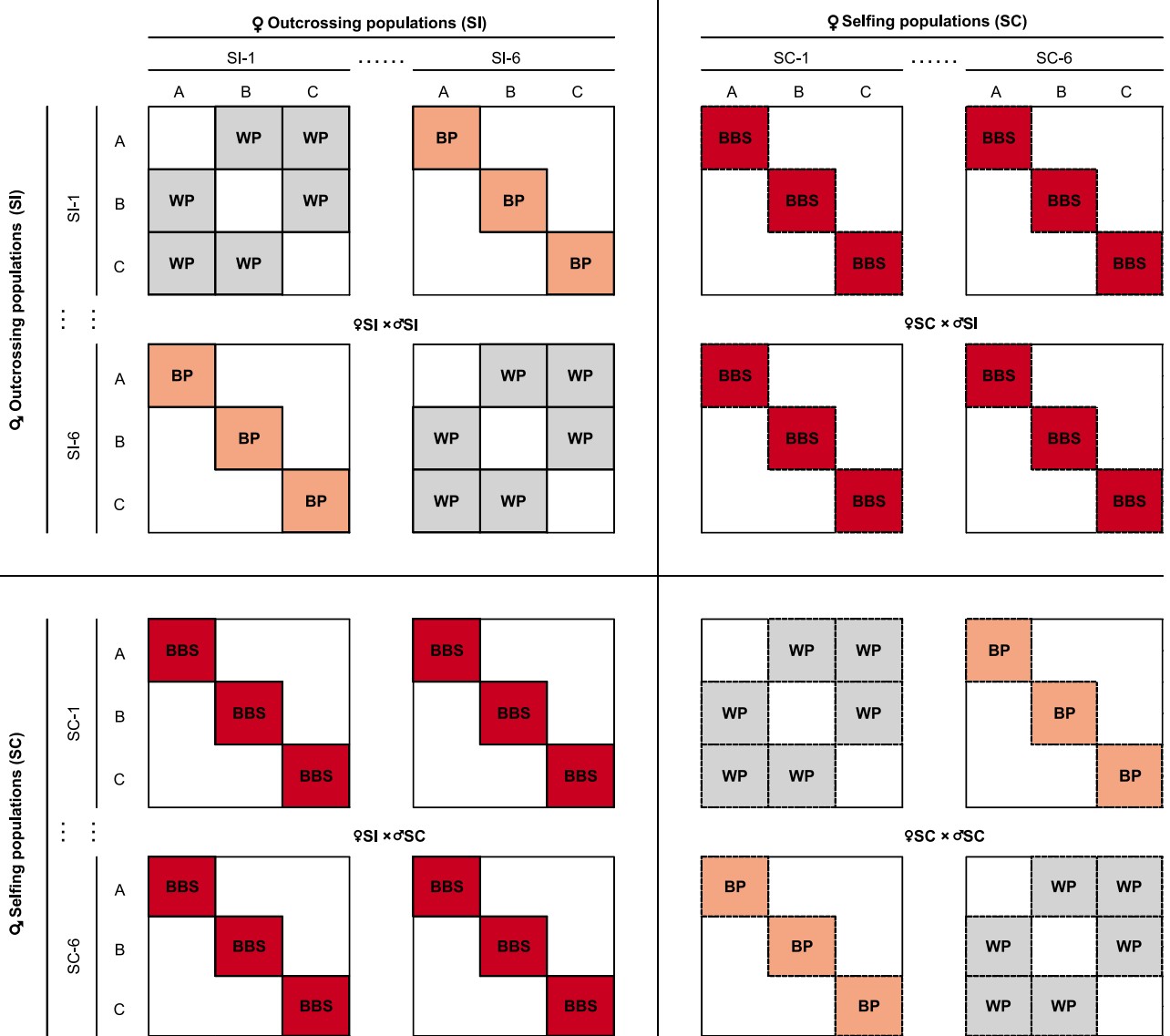

**Fig. 4 | Schematic representation of the crossing design.** We generated within-population (WP, grey), between-population (BP, orange; always within a breeding system) and between-breeding-system (BBS, red; always between populations) crosses using three parents (A, B, C) from six outcrossing (SI) populations (SI-1 to SI-6) and six selfing (SC) populations (SC-1 to SC-6). Solid borders indicate crosses where the maternal parent was SI, dashed borders where the maternal parent was SC. We duplicated the design with three further plants (D, E, F; not shown in this scheme) for each population.

population TSS[28,47], and displays *S*-allele diversity[29], perhaps reflecting a mixed mating system. Two SC populations (RON and PTP) are closely related to each other based on genetic clustering analyses[28,47], and consist of homozygotes for *S*-locus allele 1 ($S_1$)[29]. The remaining three SC populations consist of homozygotes for *S*-locus allele 19 ($S_{19}$)[29], but belong to separate genetic clusters. Matching its geographic position, LPT clusters separately, but not too distantly from the RON-PTP cluster. TC clusters with the nearby TSS and TSSA population. KTT forms a separate cluster, in line with its geographic isolation[28].

In March 2014, to test whether self-incompatibility could be restored through complementation, we randomly selected 18 focal plants (three plants labelled A, B and C from each of the six SC populations; Fig. 4). We used the selected parental SC plants to produce progeny through between-population crosses (BP$_{♀SC × ♂SC}$) among all plants labelled A, and the same for all plants labelled B and C. As controls, we made crosses within SC populations (WP$_{♀SC × ♂SC}$) among the A, B and C plants, respectively. To test whether progeny from crosses between SC and SI plants would show variation in breeding system, we also randomly selected 18 SI focal plants (three

plants labelled A, B and C from each of the six SI populations; Fig. 4). We used these to produce progeny through crosses between breeding systems (BBS$_{♀SI × ♂SC}$ and BBS$_{♀SC × ♂SI}$) among all SI and SC focal plants labelled A, and the same for the plants labelled B and C. As controls for SI populations, we made crosses within SI populations (WP$_{♀SI × ♂SI}$) among the A, B and C plants, respectively. To test whether SC plants can arise when crossing more distantly related SI plants, we also made crosses between SI populations (BP$_{♀SI × ♂SI}$) among all plants labelled A, and the same for the plants labelled B and C (Fig. 4). In August 2014, we replicated the complete crossing design (as summarized in Fig. 4 for the A, B, and C parental plants) with 36 different parental focal plants (three plants labelled D, E and F from each of the 12 populations).

In principle, our design would have generated 13 seed families per parental focal plant: 5 from crosses with plants from populations with the same breeding system (BP$_{♀SC × ♂SC}$ or BP$_{♀SI × ♂SI}$), 6 from crosses with plants from populations with a different breeding system (BBS$_{♀SI × ♂SC}$ or BBS$_{♀SC × ♂SI}$) and 2 from crosses with plants from the same population (WP$_{♀SC × ♂SC}$ or WP$_{♀SI × ♂SI}$). In total, for the 72 (36 + 36) parental plants, the complete design would thus have

yielded 936 (468 + 468) seed families. However, not all crosses resulted in seeds, so that we obtained 898 seed families (446 + 452).

As not all populations are completely fixed for one or the other breeding system, we assessed the breeding system for all parental plants by performing at least six self-pollinations per plant on at least three different days. For the SC populations, this confirmed that all 36 plants but one (genotype D of population TSSA) were indeed SC (Supplementary Data 2). For the SI populations, plants were either completely SI or leaky SI. Following criteria set in Refs. [28,43], the phenotype leaky SI is used for cases where self-pollination led to fruits with seeds, often a variable number among replicates, but fewer seeds per fruit than after cross-pollination with compatible pollen.

To avoid accidental self-pollination when making crosses, we emasculated potential recipient flowers of SC plants in the bud-stage before anther dehiscence. As a control, we left at least one emasculated flower without cross-pollen. In rare cases where this control led to fruit development (which would indicate accidental self-pollination), we discarded any fruits from flowers on the same plant cross-pollinated on the same day. To obtain at least one developed fruit for each recipient-donor combination, we did up to three cross-pollinations per combination. If none of these resulted in fruit set, we considered the parents to be cross-incompatible. All fruits were collected when mature (4–7 weeks after pollination) and stored under dry and cool conditions until further use.

## Growth of progeny

To determine their breeding system, we sowed progeny from all 898 seed families in a growth chamber with a 16 h daylight period, keeping the temperature between 17 °C and 21 °C during the day and at 15 °C during the night, and a humidity above 50%. For practical reasons, we sowed in five batches such that one seed per seed family was sown at one time, either for the 446 families derived from the A, B and C parents (batch 1, 2 and 5), or for the 452 families from the D, E and F parents (batch 3 and 4). Additionally, to increase the sample size for between-breeding-system crosses (BBS$_{♀SI × ♂SC}$ and BBS$_{♀SC × ♂SI}$), for this particular cross-type, we sowed seeds from all parental plants (A-F) in a sixth batch, and from parental plants A-C in a seventh and eighth batch. Batches were sown between July 2015 and February 2018 (Supplementary Table 4), by placing single seeds on moistened peat-based nutrient-poor substrate (Einheitserde und Humuswerke Gebr. Patzer GmbH & Co., Sinntal, Germany) randomly assigned to a 2.5 cm ×3.2 cm × 11.0 cm cell in 54-cell QuickPot trays (QP 54 T/11, Herkuplast Kubern GmbH, Ering/Inn, Germany). Seedlings were transplanted into individual square polypropylene pots (7 cm × 7 cm × 6.5 cm, Pöppelmann, Germany) filled with the same substrate as for germination. We watered twice a week and fertilized weekly with 50 ml of 0.1% Scotts Universol® blue solution (Everris International B. V., Waardenburg, Netherlands).

## Determination of the breeding system

Although not all seeds in each batch germinated and some plants did not flower within a few months from sowing, we could perform self-pollinations on a total of 1,603 progeny (Supplementary Table 4) from 653 out of 898 seed families representing all 144 (12×12) population combinations. On each plant that flowered, we aimed to perform at least ten self-pollinations on at least five different days by rubbing a ripe anther from a donor plant on the stigma of the recipient flower. This applied to 1,530 plants: 1,385 with the desired 10 or more self-pollinations and an additional 145 with at least five self-pollinations. Plants with fewer than five self-pollinations were excluded.

After pollination, fruits elongate to accommodate the developing seeds, and attain their final length one to two weeks after pollination[43,48]. Fruit length is a good proxy of seed number (Supplementary Fig. 4). Therefore, as seeds can only be counted reliably at least four weeks after pollination, we used fruit length at two weeks as a

proxy of seed set to enable a higher throughput and allow screening of more plants. Fruits without any developing seeds do not elongate and thus roughly maintain the length of the ovary (FL$_{zero}$), whereas fruits with full seed set elongate to a maximum fruit length (FL$_{max}$). Values of FL$_{zero}$ and FL$_{max}$ might vary among individual plants. Therefore, after pollination and fertilization, we measured the fruit lengths at least two weeks after self-pollination, and subsequently expressed the degree of self-compatibility relative to FL$_{zero}$ and FL$_{max}$. As it was logistically not feasible to perform control emasculations (to obtain FL$_{zero}$ directly) and reference outcrosses (to obtain FL$_{max}$ directly) for each progeny in the design, we used the available information from the parental plants to calculate expected FL$_{zero}$ and FL$_{max}$ values for each progeny. As there is considerable within and among-population variation in the maximum fruit lengths, we first calculated FL$_{max}$ for each parent individually by calculating the median fruit length for each of the four used pollen donor types (self, within population, between population, between breeding system), and taking the maximum of these medians:

$$FL_{parent_{max}} = max(median(FL_{self}), median(FL_{WP}), median(FL_{BP}), median(FL_{BBS}))$$
(1)

Then, given that progeny fruit length is inherited additively (Supplementary Table 5), we calculated the average expected FL$_{max}$ for each progeny:

$$FL_{progeny_{max}} = \frac{1}{2}\left(FL_{♀parent_{max}} + FL_{♂parent_{max}}\right)$$
(2)

There is limited variation for the length of fruits without developing seeds within populations, but there are differences among populations[44]. To account for this, we calculated a population-specific FL$_{zero}$ as twice the mean pistil length in non-pollinated flowers for each population reported in[44]. Then, again assuming an additive contribution of the maternal and paternal parent of each progeny, we calculated the average expected FL$_{zero}$ fruit length for each progeny:

$$FL_{progeny_{zero}} = \frac{1}{2}\left(FL_{♀maternal\,population_{zero}} + FL_{♂paternal\,population_{zero}}\right)$$
(3)

Based on these and the average fruit length resulting from self-pollination, we calculated an index of self-compatibility (SC) for each progeny:

$$SC\text{-}index = \frac{FL_{progeny} - FL_{progeny_{zero}}}{FL_{progeny_{max}} - FL_{progeny_{zero}}}$$
(4)

In principle, this index ranges from 0 (complete self-incompatibility) to 1 (complete self-compatibility), but is not mathematically bound by 0 and 1 due to variation in the estimates of the different component parameters. Conservatively, we considered plants with an SC-index < 0.25 as SI, and plants with an SC-index > 0.75 as SC.

Progeny that do not produce any elongated fruits (SC-index = 0) after self-pollination are considered to be SI, although in principle female or male sterility would give a similar outcome. To discern truly SI progeny from ones with sterility, the female and male fertility of all apparent SI progeny was tested by using them as donor (to test male fertility) and recipient (to test female fertility) in crosses with up to two haphazardly chosen unrelated progeny (only SI progeny were used if testing male fertility of the progeny). This was done for all 358 progeny that appeared SI in the first four batches, and revealed 27 progeny that did not produce seeds with either partner. Such progeny are not necessarily sterile, as cross-incompatibility could also explain lack of seed formation after outcrossing, but we conservatively excluded the 27 potentially sterile progeny from further analyses (see Supplementary Table 4). Thus, our final dataset included 1503 plants (1530−27).

## Inferring *S*-locus genotypes

To test whether the association between self-compatibility and the $S_1$ and $S_{19}$ haplotypes has a functional basis, we determined the *S*-locus genotype of $F_1$ progeny of crosses between SI plants and SC plants from the $S_1$ and $S_{19}$ backgrounds that displayed variation in breeding system (mixtures of SC and SI progeny). The few available allele-specific primers for *S*-locus genotyping co-amplify unlinked genes encoding other members of the receptor kinase family. This, and high divergence and length variation between *S*-alleles generally impede directly inferring *S*-locus genotypes by PCR-based methods[43,49–51]. Therefore, we inferred *S*-locus genotypes of parents and progeny indirectly by genotyping the gene B80, which is linked to the *S*-locus, but plays no direct role in self-incompatibility. In contrast to *S*-alleles, B80 can be PCR-amplified reliably and has no length variation, thus allowing direct sequencing of PCR amplicons without cloning. Based on known linkages of B80 haplotypes to certain *S*-alleles[52], B80 sequence information can then be used for indirect *S*-locus genotyping[29]. In brief, we extracted DNA from silica-dried leaf material using the E.Z.N.A.® Plant DNA Kit (Omega Bio-tek, Norcross, USA) following the manufacturer's recommendations. To amplify B80, we used 1 pmol forward (5′-GAATC- AGCAGCTTCAACCAAA-3′) and 1 pmol reverse primer (5′-GTTATCCTCCAATCGGGTCATAC-3′)[53] in 25 µl PCR mixtures further containing 0.5 U *Taq* polymerase (DreamTaq DNA Polymerase, Thermo Scientific), 1 × *Taq* polymerase buffer (DreamTaq Buffer, Thermo Scientific), 200 µM dNTPs, 2.5 mM MgCl₂. PCR-Amplification was carried out in a T-Professional Basic 96 Gradient Thermocycler (Analytik Jena, Jena, Germany) with initial denaturation at 94 °C for 3 min, annealing at 62 °C for 1 min and a final extension at 72 °C for 2 min, followed by 34 cycles of 94 °C for 30 s, 62 °C for 30 s and 72 °C for 1 min; with a final extension at 72 °C for 6 min (derived and optimized from Haudry et al. [52]). To prepare samples for direct sequencing, we purified PCR products by incubating them with 1.6 U FastAP (Thermosensitive Alkaline Phosphatase, Thermo Scientific) and 16 U Exo I (Exonuclease I, Thermo Scientific) at 37 °C for 15 min with a consecutive inactivation step at 85 °C for 15 min. Sanger sequencing was outsourced to Eurofins Genomics, Konstanz, Germany. We visually checked the ABI files for calling errors and heterozygous positions, manually adding IUPAC ambiguity codes for heterozygous sites and ambiguous base calls. To identify haplotypes, we then aligned the edited sequences to the B80-sequence library from Mable et al. [29] (https://doi.org/10.5061/dryad.832t8) using the Muscle algorithm[54] implemented in MEGA11[55] (version 11). We then compared all ambiguous calls in each sequence to the 377 reference sequences, in order to evaluate whether it likely represented a true SNP or not. We considered ambiguous calls to be true heterozygous SNPs if they were on positions that were polymorphic within the reference library. Alternatively, if they were on positions that were monomorphic within the reference library, we assumed the ambiguity was due to noise and set the base-call to the conserved state. To identify B80-haplotypes, we searched for the closest matches in the reference library. If this yielded more than two matches, we treated all matching haplotypes as candidates and finalized B80-haplotype identification based on inheritance patterns in the $F_1$ progeny. Finally, we assigned putative *S*-genotypes based on known linkages between B80 variants and *S*-alleles following Mable et al. [29]. As further confirmation of putative *S*-genotypes, we used *S*-allele-specific primers for $S_1$, $S_3$, $S_{13}$, $S_{19}$, $S_{20}$ and $S_{39}$ (see Supplementary Table 6 for details).

## Statistical analyses

To test for differences in SC-index between cross-types, we used linear mixed effects models implemented in the *lme* function of the 'nlme' package in R 4.2.3[56,57]. With the SC-index as the dependent variable, the model fixed part included cross-type (WP$_{♀SI × ♂SI}$, WP$_{♀SC × ♂SC}$, BP$_{♀SI × ♂SI}$, BP$_{♀SC × ♂SC}$, BBS$_{♀SI × ♂SC}$ and BBS$_{♀SC × ♂SI}$), and the model random part included maternal population, maternal individual (nested in

maternal population), paternal population, paternal individual (nested in parental population) and batch number. To enable using a Gaussian error distribution and ensure an appropriate normality and homogeneity of model residuals, we transformed the SC-index, which ranged from −0.39 to 2.71, by adding 1.39 to all values and subsequent natural log-transformation. To account for heterogeneity of variance (i.e. differences in variance between the different cross-types), the model included a VarIdent variance structure that allowed each level of the factor cross type to have a different variance[58].

To compare the mean SC-index among the different cross-types, we specified a matrix defining a series of 13 custom linear comparisons between different cross-type combinations (contrasts C1 to C13) and used the *glht* function in the *multcomp* package[59] to perform z-tests corrected for multiple comparisons (Supplementary Table 1). First, we tested whether progeny from between- and within-population crosses had a different SC-index, both for SI populations (C1: BP$_{♀SI × ♂SI}$ vs. WP$_{♀SI × ♂SI}$) and for SC populations (C2: BP$_{♀SC × ♂SC}$ vs. WP$_{♀SC × ♂SC}$). Second, we tested whether the SC-index of progeny from between-population crosses differed between SC and SI populations (C3: BP$_{♀SI × ♂SI}$ vs. BP$_{♀SC × ♂SC}$). Third, we tested whether the SC-index of progeny from between-breeding-system crosses depended on the direction of the cross (C4: BBS$_{♀SI × ♂SC}$ vs. BBS$_{♀SC × ♂SI}$). When C4 was not significant, we merged BBS$_{♀SI × ♂SC}$ and BBS$_{♀SC × ♂SI}$, and did further contrasts to test for phenotypic additivity (C5: BBS vs. BP), complete phenotypic dominance of self-incompatibility (C6: BBS vs. BP$_{♀SI × ♂SI}$) or complete phenotypic dominance of self-compatibility (C7: BBS vs. BP$_{♀SC × ♂SC}$). When C4 was significant, we did not merge BBS$_{♀SI × ♂SC}$ and BBS$_{♀SC × ♂SI}$ and used separate tests for phenotypic additivity (C8: BBS$_{♀SI × ♂SC}$ vs. BP and C9: BBS$_{♀SC × ♂SI}$ vs. BP), phenotypic dominance of self-incompatibility (C10: BBS$_{♀SI × ♂SC}$ vs. BP$_{♀SI × ♂SI}$ and C11: BBS$_{♀SC × ♂SI}$ vs. BP$_{♀SI × ♂SI}$,), and phenotypic dominance of self-compatibility (C12: BBS$_{♀SI × ♂SC}$ vs. BP$_{♀SC × ♂SC}$ and C13: BBS$_{♀SC × ♂SI}$ vs. BP$_{♀SC × ♂SC}$).

## Reporting summary

Further information on research design is available in the Nature Portfolio Reporting Summary linked to this article.

## Data availability

The B80 gene sequence data generated in this study has been deposited in the NCBI Genbank sequence database under accession OQ798221-OQ798795. The underlying raw sequences (*.ab1 format) and B80 reference library are available on Figshare [https://doi.org/10.6084/m9.figshare.22439239]. Source data is provided with this paper. Raw data are available from Figshare: https://doi.org/10.6084/m9.figshare.22439257.

## Code availability

All R code is available from Figshare [https://doi.org/10.6084/m9.figshare.22439257].

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

## Acknowledgements
We thank Barbara Mable for providing seeds from natural populations, Sina Konitzer-Glöckner for help with crossing, and Heinz Vahlenkamp, Karoline Jetter, Yanjie Liu, Samuel Fernandes and Beate Rüter for help with the experiment. YL was funded by a scholarship from the China Scholarship Council.

## Author contributions
M.S. conceived the study. M.S. and Mv.K. designed the experiment. Y.L. and N.K. executed the experiment. N.K. and E.M. performed molecular work (B80 and S-locus genotyping) with input from M.S. Y.L. analyzed the data with input from M.S. and Mv.K. Y.L. and M.S. wrote the paper with input from Mv.K.

## Funding

## Competing interests
The authors declare no competing interests
