## [Peer Review File · Nature Communications]

Breakdown of self-incompatibility due to genetic interaction between a specific S-allele and an unlinked modifierReviewers' Comments:

Reviewer #1:

Remarks to the Author:

Li, Van Kleunen & Stift provide a nice study based on crossing-experiments to unravel the genetic architecture of the "loss of self incompatibility" in an otherwise outcrossing species (namely *Arabidopsis lyrata*). It has been demonstrated earlier by various authors that in particular in the Great Lakes system in the northern US *Arabidopsis lyrata* exists with self-compatible populations. The N American gene pool is derived, since the species is of a Pleistocenic and Central European origin. A key finding is that specific S-locus haplotypes may cause SC simply by interacting with one/few unlinked modifiers.

In general I see that statistical analyses have been conducted in a proper way and results have been carefully discussed in the given context.

However, I do see quite a number of limitations and open questions that I feel are important to be answered, considered and/or addressed.

1) With the introduction (pg. 2, line 22ff.): I cannot follow the argument linking "taxonomy" (sister species) with "secondary mutations". This is all a matter of time (and space) and respective separation of gene pools. Indeed, accumulation of such "secondary mutations" coincides often with taxonomical treatment, but it must not. *Arabidopsis lyrata* itself is a nice example. American *A. lyrata* is treated as subspecies *lyrata* and Eurasian *A. lyrata* is most often recognized as subspecies *petraea*. However, in arctic N America there is also self-compatible *Arabidopsis arenicola*, which may have derived postglacially from nearby *A. lyrata* ssp. *lyrata*. This little example should demonstrate that evolutionary history largely matters, WHICH HAS BEEN FULLY IGNORED IN THIS CONTRIBUTION.

2) The history of populations has not been analysed. Buckley et al. (2018, BMC Genomics) curiously enough presented a "population dendrogram" based on radSeq data (exactly the same populations) placing three of the herein analysed SC-populations as ancestral to the remainder. The two other SC populations are nested within SI populations. It seems that the data are influenced by strong biogeographical and "phylogenetic" (population history) signal and a number of questions may arise that are also important for the given study: Past and present gene flow? Mixed mating systems within populations? Etc. Btw: The study of Buckley et al. has not been cited at all.

3) I am also missing some comments on the timing of the scenario worked on. *Arabidopsis lyrata* might have colonized America 190-310,000 years ago (Mattila et al. 2017) and thereby must have undergone a complex Glaciation/Deglaciation history with severe isolation, bottlenecks, secondary contact etc. This may have allowed to accumulate exactly those "secondary mutations" the authors aimed to avoid by choosing the system

4) Considering #1-3 it is not surprising to see that the hypothesized modifier loci and its (dominant) alleles remain hidden. The key question remains: Why does this happen in N America and not in Eurasia, with somehow similar phylogeographic scenarios? Is SC *A. arenicola* in arctic Canada/Alaska a simple postglacial "escape" from SC *A. lyrata* ssp. *lyrata*? Why was this "species" not included?

Reviewer #2:

Remarks to the Author:

Identifying the mutations at the origin of the breakdown of self-incompatibility (hereafter SI) is particularly important to understand the molecular basis underlying the emergence of self-compatible (SC) genotypes and how they may segregate and evolve within populations. In particular, the major challenge in studying the loss of SI is distinguishing primary inactivating mutations from secondary ones.

In this manuscript, Li and colleagues characterized the breeding system inheritance in north American *Arabidopsis lyrata* populations with contrasted breeding systems (outcrossing or selfing). Based on the frequencies of SC and SI genotypes produced by intra and inter population crosses, the authors proposed a two-locus genetic model which involves a modifier locus interacting with S1 or S19 S-haplotypes generally associated with SC in these populations.

This is a very well written and well performed study and I enjoyed reading this paper. The authors performed an impressive crossing experiment which has been particularly well designed to test for different hypothesis for genetics models underlying the emergence of SC genotypes. The methods to characterize the breeding system of the progeny is conservative and the statistical methods to test genetic models look robust. The figures are also well chosen to summarize the results.

I have a couple of minor comments or suggestions.

1- I wonder if we could have an effect of the functional S-haplotypes in SI populations (and their dominance levels), on the expression of SC phenotypes in the progeny of SIxSC crosses, and depending on the dominance level of the allele associated with SC (S1 and S19)? It could be developed for the discussion.

2-Lines 321: "Our first main finding is that most progeny from crosses between SC parents from six different selfing populations were also SC. This shows that the loss of self-incompatibility in the different selfing populations of *A. lyrata* cannot be explained by different recessive loss-of-function mutations."

In F1, we do not necessarily expect complementation for different loss of function mutations if they are located in the same gene (therefore leading each to nonfunctional proteins).

Reviewer #3:

Remarks to the Author:

In this manuscript, Li and colleagues present the results of a large crossing experiment within and between North American populations of *Arabidopsis lyrata* that differ in their breeding system. The main aim of the study is obtaining an improved understanding of the genetic basis of loss of self-incompatibility in *A. lyrata*. They interpret their findings to suggest that loss of self-incompatibility in *A. lyrata* has involved mutations at two loci, involving a combination of one of two S-haplotypes and different unlinked modifier loci. The foundation for this conclusion is statistical analysis of the outcome of a large set of crosses.

Overall, I am not convinced that this manuscript is suitable for a journal with a broad readership, as the conclusions from the study do not go beyond the model system itself and the study does not test broader theoretical predictions of general importance. The methods used are also not novel or original in relation to other work in the field. While I appreciate the large amount of work that this study entailed, I have concerns about the experimental setup and lack of direct genetic validation of the inferred model.

Major concerns

1. Lack of direct validation of genetic model

The main aim of this study is to investigate the genetic basis of loss of SI in *A. lyrata*. As this is the main aim, I wonder why the authors did not attempt to use genetic data to directly validate their inferred genetic model (e.g. using genetic mapping in F2 mapping populations from crosses of SI and SC populations). Incorporating a direct validation of their genetic model based on genetic mapping would make the study more broadly interesting, provide a basis for further genetic work aiming at identifying causal mutations, and should be feasible, given that the cost of genotyping mapping populations at thousands of loci has been drastically reduced recently. This is something I would definitely have expected to see in a study on this topic targeted for publication in a journal with a broad readership.

2. Lack of validation of inference of self-incompatibility/self-compatibility

The inference of genetic self-incompatibility (SI) vs self-compatibility (SC) is done solely based on fruit length, without validation using direct assessment of pollen tube growth in even a subset of individuals. That also means that any early-acting inbreeding depression or incompatibilities between populations can potentially affect the findings. If such validation data are available, they should be presented in the Supplementary and referred to in the manuscript, otherwise potential biases should be discussed in greater depth.

3. Results are not placed in a broader context, presentation could be improved

The authors could better explain why it is of interest to understand the genetic basis of loss of self-incompatibility, for instance by more comprehensively referring to previous work. There is a rich body of theoretical work that has outlined expectations for the contribution of different types of mutations, at the S-locus and at unlinked loci, and it would be nice if the authors would place their work in this broader context. On this note, some of the text in the Introduction and Discussion does not accurately represent earlier work. The authors claim that there is no evidence that mutations at the S-locus are responsible for loss of self-incompatibility in *Capsella rubella* (line 59), but do not cite previous work that mapped the genetic basis of loss of SI to the S-locus (Sicard et al. 2011, Slotte et al. 2012) and that make it particularly meaningful to analyze S-locus sequences to identify causal mutations. The authors also do not cite relevant work on other Brassicaceae species such as *Arabidopsis kamchatica*, *Capsella orientalis* and *Leavenworthia alabamica*.

One of the main motivations for the present study is that there is intraspecific variability in SI in *A. lyrata*, and the authors claim that this will make it easier to identify causal mutations (e.g. lines 356-359) as there will be less scope for secondary mutations if the loss was recent. I am not sure that this claim is justified - to support it, the authors should elaborate more on what is known about the timing of loss of SI in *A. lyrata* in relation to the timing of loss of SI in the other systems that have been studied, and cite work that is relevant to this question. While Foxe et al. (2010) have investigated this question using microsatellites, there is also more recent sequence-based work that might be relevant. Is it the case that we can be confident that SI was lost much more recently in *A. lyrata* than in the other crucifer systems studied so far, some of which have very recent divergence (and in the case of *Leavenworthia alabamica* also intraspecific divergence in mating system between populations)?

Finally, the presentation could be improved. While the Introduction and Discussion is overall well-written and easy to read, the Material and Methods and Results are somewhat difficult to read, and Table 2 is difficult to parse.

4. Model assumptions and interpretation

Interpretation of the results from this study rests heavily on some restrictive assumptions. For instance, self-incompatibility (SI) would only be expected to be restored by complementation in crosses between self-compatible populations if the populations harbored two different recessive loss-of-function mutations. This means that if SI is not restored, it could be due to just one of the populations breaking this specific assumption, for instance by harboring a dominant mutation leading to SC. Indeed, the authors conclude that mutations with dominant effects seem to be involved in the loss of SI in *A. lyrata*. Thus, the use of the complementation approach to study the breakdown of SI is perhaps not ideal in general. Indeed, dominant loss-of-function SC mutations are known to occur at the S-locus and have been documented in e.g. *Arabidopsis kamchatica*, *Capsella rubella*, and more recently in *Capsella orientalis*. Thus, assuming that loss-of-function mutations will be recessive at the S-locus just as we may expect at other loci seems overly simplistic. On the contrary, it is not surprising that dominant loss-of-function mutations could occur specifically at the S-locus, as it is well known that there is a dominance hierarchy of S-alleles, with pollen-level dominance mediated by trans-acting S-linked sRNAs expressed by dominant S-alleles, that transcriptionally silence recessive S-alleles. This part of the discussion should be modified to better reflect that expectations for the S-locus differ from those at other loci in the genome (lines 333-340).

Minor comments

Figure 1, please correct typo in figure "Outcrossing"

Line 43, "pollen tubes cannot grow and preclude fertilization", please rephrase to "pollen tubes cannot grow and this prevents fertilization"

Abstract: "As the first step towards the evolution of selfing from obligate outcrossing, identifying the key mutations underlying the loss of self-incompatibility is of particular interest." Please rephrase to clarify the meaning of this sentence.

Reviewer #1 (Remarks to the Author):

Li, Van Kleunen & Stift provide a nice study based on crossing-experiments to unravel the genetic architecture of the “loss of self incompatibility” in an otherwise outcrossing species (namely *Arabidopsis lyrata*). It has been demonstrated earlier by various authors that in particular in the Great Lakes system in the northern US *Arabidopsis lyrata* exists with self-compatible populations. The N American genepool is derived, since the species is of a Pleistocenic and Central European origin. A key finding is that specific S-locus haplotypes may cause SC simply by interacting with one/few unlinked modifiers.

In general I see that statistical analyses have been conducted in a proper way and results have been carefully discussed in the given context. However, I do see quite a number of limitations and open questions that I feel are important to be answered, considered and/or addressed.

1) With the introduction (pg. 2, line 22ff.): I cannot follow the argument linking “taxonomy” (sister species) with “secondary mutations”. This is all a matter of time (and space) and respective separation of genepools. Indeed, accumulation of such “secondary mutations” coincides often with taxonomical treatment, but it must not. *Arabidopsis lyrata* itself is a nice example. American *A. lyrata* is treated as subspecies *lyrata* and Eurasian *A. lyrata* is most often recognized as subspecies *petraea*. However, in arctic N America there is also self-compatible *Arabidopsis arenicola*, which may have derived postglacially from nearby *A. lyrata* ssp. *lyrata*. This little example should demonstrate that evolutionary history largely matters, WHICH HAS BEEN FULLY IGNORED IN THIS CONTRIBUTION.

*We fully agree that evolutionary history is important and relevant as background information, and we have now included more evolutionary/biogeographical background. In the introduction, we have now also added a reference to the *A. arenicola* example mentioned by the reviewer (L69 - 70). Furthermore, in the methods-section on source plant material, we expanded the section giving background information about the study species (L103 -116). Finally, since we had no intention to suggest that secondary mutations are impossible in *A. lyrata*, to avoid any confusion, we no longer include this argumentation.*

2) The history of populations has not been analysed. Buckley et al. (2018, BMC Genomics) curiously enough presented a “population dendrogram” based on radSeq data (exactly the same populations) placing three of the herein analysed SC-populations as ancestral to the remainder. The two other SC populations are nested within SI populations. It seems that the data are influenced by strong biogeographical and “phylogenetic” (population history) signal and a number of questions may arise that are also important for the given study: Past and present geneflow? Mixed mating systems within populations? Etc. Btw: The study of Buckley et al. has not been cited at all.

We are not sure we understand the comments correctly, but in our opinion the dendrogram in Buckley et al cannot be interpreted as evidence for the ancestral state of LPT, RON and PTP. The dendrogram merely shows that the three SC populations LPT, RON and PTP cluster separately from the other study populations (including the other SC populations). This does, however, not support that they are ancestral, just that they have a common ancestor to which the LPT-RON-PTP clade is a bit closer. Also, more importantly (we think), it does not mean that the common ancestor is SC. We do agree that the insights from Buckley et al 2018 are important for our interpretations. In hindsight, our presentation may have given the

wrong impression that selfing populations fixed for S19 are closely-related to each other. Clearly, they are not, even if they carry the same S-locus variant. We now clarify this better in the Methods (Source plant information), where we also point out more explicitly the potential mixed-mating status of TSSA (L106 -116).

3) I am also missing some comments on the timing of the scenario worked on. *Arabidopsis lyrata* might have colonized America 190-310.000 years ago (Mattila et al. 2017) and thereby must have underwent a complex Glaciation/Deglaciation history with severe isolation, bottlenecks, secondary contact etc. This may have allowed to accumulate exactly those “secondary mutations” the authors aimed to avoid by choosing the system

We agree that our data cannot say anything about the timing of the breakdown of self-incompatibility, and thus we no longer include reference to primary/secondary mutations.

4) Considering #1-3 it is not surprising to see that the hypothesized modifier loci and its (dominant) alleles remain hidden. The key question remains: Why does this happened in N America and not in Eurasia, with somehow similar phylogeographic scenarios? Is SC A. *arenicola* in arctic Canada/Alasca a simple postglacial “escape” from SC A. *lyrata* ssp. *lyrata*? Why was this “species” not included?

We agree that it remains unresolved why SC evolved in North America, and not in Europe (at least, no SC populations have been discovered there so far). However, it was not the goal of this study to resolve this intriguing question. To avoid the impression that it was, we have now set up the introduction differently, clarifying that our main goal was to test whether the cause of the breakdown of self-incompatibility was likely due to S-locus mutations or may have had a cause outside of the S-locus (L64 - 66; L74 - 77; L82 - 94).

Including A. arenicola of course would have been interesting, but beyond the scope of our study.

Reviewer #2 (Remarks to the Author):

Identifying the mutations at the origin of the breakdown of self-incompatibility (hereafter SI) is particularly important to understand the molecular basis underlying the emergence of self-compatible (SC) genotypes and how they may segregate and evolve within populations. In particular, the major challenge in studying the loss of SI is distinguishing primary inactivating mutations from secondary ones.

In this manuscript, Li and colleagues characterized the breeding system inheritance in north American *Arabidopsis lyrata* populations with contrasted breeding systems (outcrossing or selfing). Based on the frequencies of SC and SI genotypes produced by intra and inter population crosses, the authors proposed a two-locus genetic model which involves a modifier locus interacting with S1 or S19 S-haplotypes generally associated with SC in these populations.

This is a very well written and well performed study and I enjoyed reading this paper. The authors performed an impressive crossing experiment which has been particularly well designed to test for different hypothesis for genetics models underlying the emergence of SC genotypes. The methods to characterize the breeding system of the progeny is conservative

and the statistical methods to test genetic models look robust. The figures are also well chosen to summarize the results.

We thank the reviewer for the very positive assessment.

I have a couple of minor comments or suggestions.

1) I wonder if we could have an effect of the functional S-haplotypes in SI populations (and their dominance levels), on the expression of SC phenotypes in the progeny of SIxSC crosses, and depending on the dominance level of the allele associated with SC (S1 and S19)? It could be developed for the discussion.

This is a very good point, and the reviewer was spot on. We now included B80 haplotypes (to infer S-locus genotypes) to address this. Indeed, it turned out that S-allele dominance levels play a critical role in explaining the patterns of SI and SC in the cross-progeny. We have integrated the inference of S-locus genotypes into the methods and results and added the interpretation to the discussion (L396 - 398; L424 - 437).

2) Lines 321: “Our first main finding is that most progeny from crosses between SC parents from six different selfing populations were also SC. This shows that the loss of self-incompatibility in the different selfing populations of *A. lyrata* cannot be explained by different recessive loss-of-function mutations.”

In F1, we do not necessarily expect complementation for different loss of function mutations if they are located in the same gene (therefore leading each to nonfunctional proteins).

We agree, and have toned down our interpretations on this matter (L372 - 373).

Reviewer #3 (Remarks to the Author):

In this manuscript, Li and colleagues present the results of a large crossing experiment within and between North American populations of *Arabidopsis lyrata* that differ in their breeding system. The main aim of the study is obtaining an improved understanding of the genetic basis of loss of self-incompatibility in *A. lyrata*. They interpret their findings to suggest that loss of self-incompatibility in *A. lyrata* has involved mutations at two loci, involving a combination of one of two S-haplotypes and different unlinked modifier loci. The foundation for this conclusion is statistical analysis of the outcome of a large set of crosses.

Overall, I am not convinced that this manuscript is suitable for a journal with a broad readership, as the conclusions from the study do not go beyond the model system itself and the study does not test broader theoretical predictions of general importance. The methods used are also not novel or original in relation to other work in the field. While I appreciate the large amount of work that this study entailed, I have concerns about the experimental setup and lack of direct genetic validation of the inferred model.

The current knowledge about the breakdown of self-incompatibility is based on studies on a few other model systems. Our study is the first one that provides strong support for a novel causal mechanism for the breakdown of self-incompatibility. Therefore, we believe that our findings, especially with the newly added genetic validation (see below), are of general importance and have implications that go beyond our particular model system.

Major concerns:

1) Lack of direct validation of genetic model.

The main aim of this study is to investigate the genetic basis of loss of SI in *A. lyrata*. As this is the main aim, I wonder why the authors did not attempt to use genetic data to directly validate their inferred genetic model (e.g. using genetic mapping in F₂ mapping populations from crosses of SI and SC populations). Incorporating a direct validation of their genetic model based on genetic mapping would make the study more broadly interesting, provide a basis for further genetic work aiming at identifying causal mutations, and should be feasible, given that the cost of genotyping mapping populations at thousands of loci has been drastically reduced recently. This is something I would definitely have expected to see in a study on this topic targeted for publication in a journal with a broad readership.

We now added genetic validation of our genetic model. Specifically, we incorporated S-locus genotype information based on B80 sequencing (Fig. 4; Fig. S3; Fig S4; Table S5; Table S8).

With these additional data, our results provide much more convincing evidence that the loss of SI cannot be due to S-locus loss-of-function mutations. Instead, it must be caused by an interaction of the S-locus and an unlinked locus. As this is the first case where the S-locus is only indirectly involved in the loss of self-incompatibility, this has broad implications for our understanding of breeding and mating system evolution in general. The addition of genetic data on S-locus genotypes for the F₁ progeny genetically validates a key aspect of our model for the breakdown of SI, and a strong basis for further genetic work aiming at identifying the causal mutations. Thus, we believe our findings warrant publication in a journal with a broad readership.

2) Lack of validation of inference of self-incompatibility/self-compatibility.

The inference of genetic self-incompatibility (SI) vs self-compatibility (SC) is done solely based on fruit length, without validation using direct assessment of pollen tube growth in even a subset of individuals. That also means that any early-acting inbreeding depression or incompatibilities between populations can potentially affect the findings. If such validation data are available, they should be presented in the Supplementary and referred to in the manuscript, otherwise potential biases should be discussed in greater depth.

It is unclear to us how our main finding could be biased by early acting inbreeding depression. There is indeed a potential for early acting inbreeding depression or incompatibilities between populations to affect the findings, in the sense that plants that are in reality SC may appear more SI (or not fully SC), if early acting inbreeding depression reduces seed set or seed size and thereby suppresses fruit elongation. However, as outlined in L216 - 219 and Table S3, we verified for all 358 progeny scored as self-incompatible in the first four progeny batches whether female or male sterility could explain the lack of fruit development. Only for a very small number (27 out of 358), we could not rule out fertility issues. Additionally, fruit length is highly correlated with seed production (Fig. S1). Moreover, unless inbreeding depression is caused by a single major-effect locus, one would then expect family-specific unimodal distributions around the mean (depressed) fruit length, which does not reflect what we found: an overall and within-family bimodal distribution of fruit lengths (Fig. 2; Fig.3; Fig. S2). We now discuss this (L441 - 451).

3) Results are not placed in a broader context, presentation could be improved.

The authors could better explain why it is of interest to understand the genetic basis of loss of self-incompatibility, for instance by more comprehensively referring to previous work. There is a rich body of theoretical work that has outlined expectations for the contribution of different types of mutations, at the S-locus and at unlinked loci, and it would be nice if the authors would place their work in this broader context. On this note, some of the text in the Introduction and Discussion does not accurately represent earlier work. The authors claim that there is no evidence that mutations at the S-locus are responsible for loss of self-incompatibility in *Capsella rubella* (line 59), but do not cite previous work that mapped the genetic basis of loss of SI to the S-locus (Sicard et al. 2011, Slotte et al. 2012) and that make it particularly meaningful to analyze S-locus sequences to identify causal mutations.

The authors also do not cite relevant work on other Brassicaceae species such as *Arabidopsis kamchatica*, *Capsella orientalis* and *Leavenworthia alabamica*.

In our largely rewritten introduction and discussion, we have attempted to better clarify our logic and the broader context for our findings. Specifically, we now refer to the relevant work on Arabidopsis kamchatica, Capsella orientalis and Leavenworthia alabamica (L49; L55 - 59; L386 - 388; L425 - 426) and more frequently refer to earlier findings, including the previous studies that have mapped SC to the S-locus. Note, however, that some of the studies pointed out by the reviewer mapped the selfing syndrome, not self-compatibility.

4) One of the main motivations for the present study is that there is intraspecific variability in SI in *A. lyrata*, and the authors claim that this will make it easier to identify causal mutations (e.g. lines 356-359) as there will be less scope for secondary mutations if the loss was recent. I am not sure that this claim is justified - to support it, the authors should elaborate more on what is known about the timing of loss of SI in *A. lyrata* in relation to the timing of loss of SI in the other systems that have been studied, and cite work that is relevant to this question. While Foxe et al. (2010) have investigated this question using microsatellites, there is also more recent sequence-based work that might be relevant. Is it the case that we can be confident that SI was lost much more recently in *A. lyrata* than in the other crucifer systems studied so far, some of which have very recent divergence (and in the case of *Leavenworthia alabamica* also intraspecific divergence in mating system between populations)?

We no longer justify our study system as ideal for distinguishing primary from secondary mutations, and the comparison of the timing between Leavenworthia and A. lyrata is therefore no longer relevant. Instead, we focus on identifying the mechanism causing self-compatibility and the main finding that, unlike any of the previously discovered causal mechanisms, loss-of-function mutations at the S-locus did NOT cause the loss of SI.

5) Finally, the presentation could be improved. While the Introduction and Discussion is overall well-written and easy to read, the Material and Methods and Results are somewhat difficult to read, and Table 2 is difficult to parse.

Since we added substantial genetic data (Fig. 4; Fig. S3; Fig S4; Table S5; Table S8), we have completely rewritten the Results section, and parts of materials and methods (L103 - 116; L222 - 261). Table 2 is no longer included, and we think the overall readability is now better.

6) Model assumptions and interpretation

Interpretation of the results from this study rests heavily on some restrictive assumptions.

For instance, self-incompatibility (SI) would only be expected to be restored by complementation in crosses between self-compatible populations if the populations harbored two different recessive loss-of-function mutations. This means that if SI is not restored, it could be due to just one of the populations breaking this specific assumption, for instance by harboring a dominant mutation leading to SC. Indeed, the authors conclude that mutations with dominant effects seem to be involved in the loss of SI in *A. lyrata*. Thus, the use of the complementation approach to study the breakdown of SI is perhaps not ideal in general. Indeed, dominant loss-of-function SC mutations are known to occur at the S-locus and have been documented in e.g. *Arabidopsis kamchatica*, *Capsella rubella*, and more recently in *Capsella orientalis*. Thus, assuming that loss-of-function mutations will be recessive at the S-locus just as we may expect at other loci seems overly simplistic. On the contrary, it is not surprising that dominant loss-of-function mutations could occur specifically at the S-locus, as it is well known that there is a dominance hierarchy of S-alleles, with pollen-level dominance mediated by trans-acting S-linked sRNAs expressed by dominant S-alleles, that transcriptionally silence recessive S-alleles. This part of the discussion should be modified to better reflect that expectations for the S-locus differ from those at other loci in the genome (lines 333-340).

We have largely rewritten the discussion to reflect the new genetic data. This automatically required taking into account the dominance hierarchy of S-alleles, which we now discuss much more carefully (L396 - 398; L424 - 437). Note that the mentioned studies on Arabidopsis kamchatica, Capsella rubella and Capsella orientalis did not discover dominant loss-of-function SC mutations, but loss-of-function mutations at a dominant S-allele. The dominance of the allele was not disrupted.

Minor comments

Figure 1, please correct typo in figure "Outcrossing"

Done.

Line 43, "pollen tubes cannot grow and preclude fertilization", please rephrase to "pollen tubes cannot grow and this prevents fertilization"

Done (L43).

Abstract: "As the first step towards the evolution of selfing from obligate outcrossing, identifying the key mutations underlying the loss of self-incompatibility is of particular interest."

Please rephrase to clarify the meaning of this sentence.

Sentence was edited out.

Reviewers' Comments:

Reviewer #1:

Remarks to the Author:

Dear authors,

I appreciate that you commented on my various comments, and with the largely re-written version it is much clearer now that you are not going beyond your (experimentally confirmed) findings of SI breakdown describing a potentially new mechanism.

However, all my questions pointed towards a direction that would make the contribution of great interest of a broader readership.

As it stands my comments have been all addressed, basically while deleting and/or explicitly excluding these aspects.

The manuscript has been significantly improved, and from my perspective addition of genetic data turned this into a very nice contribution.

Reviewer #2:

Remarks to the Author:

The new version of the manuscript addresses all of my concerns. The authors propose two models to explain the loss of self-incompatibility in *A. lyrata* populations, which are associated with the S1 or the S19 allele at the S-locus. I think the new genotyping data have particularly strengthened the results. More generally, their findings are in agreement with the hypotheses of multiple origins of selfing, and strongly improve our knowledge about the transition to self-compatibility and the evolution of mating systems in plants.

I have a couple of minor comments or suggestions:

- The first sentence that mentions the dominance hierarchy between S-alleles is line 324 "Five SI progeny had inherited a haplotype associated with most recessive allele S1...".

I think it should be better introduced or mentioned before. References about S-alleles dominance phenotypes are also missing.

- Line 436. "This would resemble the scenario proposed for the Siberian selfing lineages of *A. lyrata*, where the dominant specificity AhS12 carried loss-of-function mutations in selfing, but not in outcrossing lineages". AhS12 refers to an *Arabidopsis halleri* S-allele, to my knowledge the corresponding/closest allele in *A. lyrata* is AIS42.

- The model with a modifier, which specifically inactivates the function of the S1 allele, is particularly interesting and the parallel with S-locus dominance modifiers, which are also allele-specific, could be mentioned in the discussion. In both systems, the most recessive allele S1 is repressed.

Reviewer #1 (Remarks to the Author)

This reviewer had no further suggestions.

Reviewer #2 (Remarks to the Author):

1) The first sentence that mentions the dominance hierarchy between S-alleles is line 324 “Five SI progeny had inherited a haplotype associated with most recessive allele S1...”.

I think it should be better introduced or mentioned before. References about S-alleles dominance phenotypes are also missing.

This is a good point, and we now mention the dominance/recessivity of the S₁ and S₁₉ alleles upon first mention in the introduction (line 78-79).

2) Line 436. “This would resemble the scenario proposed for the Siberian selfing lineages of *A. lyrata*, where the dominant specificity AhS12 carried loss-of-function mutations in selfing, but not in outcrossing lineages”. AhS12 refers to an Arabidopsis halleri S-allele, to my knowledge the corresponding/closest allele in *A. lyrata* is AIS42.

We added this information in brackets (line 252-253).

3) The model with a modifier, which specifically inactivates the function of the S1 allele, is particularly interesting and the parallel with S-locus dominance modifiers, which are also allele-specific, could be mentioned in the discussion. In both systems, the most recessive allele S1 is repressed.

We of course agree that this is particularly interesting. The parallel with S-locus dominance modifiers is indeed striking, and we now mention this in the discussion (line 225-230).